# 🦀CRAB: Assessing the Strength of Causal Relationships Between Real-World Events

**Angelika Romanou, Syrielle Montariol***, **Debjit Paul***,
**Léo Laugier, Karl Aberer, Antoine Bosselut**

EPFL
{firstname.lastname}@epfl.ch

## Abstract

Understanding narratives requires reasoning about the cause-and-effect relationships between events mentioned in the text. While existing foundation models yield impressive results in many NLP tasks requiring reasoning, it is unclear whether they understand the complexity of the underlying network of *causal relationships* of events in narratives. In this work, we present *CRAB*, a new **C**ausal **R**easoning **A**ssessment **B**enchmark designed to evaluate causal understanding of events in real-world narratives. *CRAB* contains fine-grained, contextual causality annotations for ∼ 2.7K pairs of real-world events that describe various newsworthy event timelines (*e.g.*, the acquisition of Twitter by Elon Musk). Using *CRAB*, we measure the performance of several large language models, demonstrating that most systems achieve poor performance on the task. Motivated by classical causal principles, we also analyze the causal structures of groups of events in *CRAB*, and find that models perform worse on causal reasoning when events are derived from complex causal structures compared to simple linear causal chains. We make our dataset and code available to the research community.

## Introduction

Understanding narratives requires understanding the cause-and-effect relationships between interconnected sub-events of those narratives. When reading text, humans immediately induce potential causal links between the events presented as part of a larger scenario (Grunbaum 1952; Pearl and Mackenzie 2018). For example, in Figure 1, when reading an article about the acquisition of Twitter in 2022, a reader would implicitly assign causal links between events such as E2: "Elon Musk closes 44 billion dollar deal to buy Twitter" and E3: "Twitter delists from the NYSE".

However, building accurate causal mental models of the situations depicted in narratives poses several complex challenges. First, human causality judgments are rarely binary. Instead, they fall on a spectrum depending on human perception of other mediating or confounding events (Pearl 2009). For example, in Figure 1, E2 is a mediator event for the causal relationship of E1 and E3, likely affecting the human perception of the causal relationship between E1 and

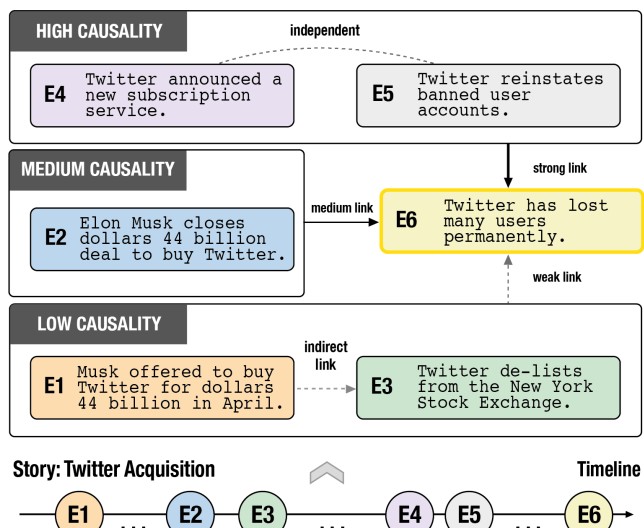

Figure 1: Events from *CRAB* that lead to event E6, forming causal sub-structures with links of various causal strength.

E3. Second, causality judgments depend on the context depicting the events in question — a context that can affect perceptions of causality. For example, a high causal judgment might be assigned between E4 and E6 in Figure 1. However, the introduction of new information about E5, another potential cause of E6, might downgrade the perceived intensity of a causal link between E4 and E6. Finally, because context is critically important to judging causal relationships of events, and most narratives offer an incomplete (and sometimes biased) reporting of particular scenarios, multiple sources may be required to paint an accurate picture of the causal relationships between multiple interconnected events.

Addressing these challenges, we introduce *CRAB*, a new **C**ausal **R**easoning **A**ssessment **B**enchmark that contains fine-grained, contextual causality annotations of real-world events that happened in the past ten years and received extensive media coverage. To collect the proposed benchmark, we design a crowdsourcing framework motivated by standard causal principles from cognitive science (Cao, Williamson, and Choi 2022) and actual causality (Halpern

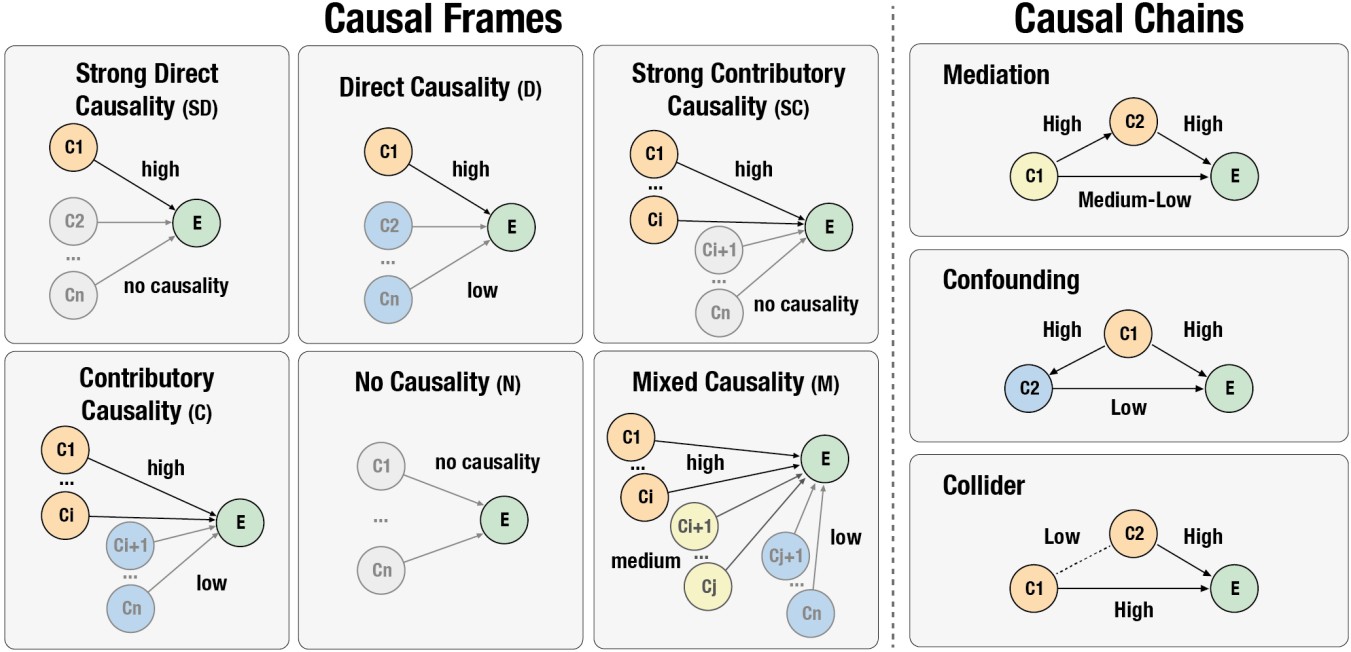

Figure 2: *Left:* Different structures of causal frames inspired by the responsibility assessment concepts presented in Halpern 2016; *right:* causal chains (Pearl 2009) present in **CRAB**. The patterns in structures are formulated based on the different causal judgment scores among events. The colors of cause-nodes represent the causality strength they have towards the event-node E.

2016) that study how humans perceive and express causality and responsibility among events. Using this knowledge frame, we automatically extract the events of newsworthy stories by integrating large pre-trained LMs into the dataset creation loop. We then construct causal graphs — combinations of inter-connected events forming different causal chains and frames, as presented in Figure 2 — from the extracted events and assess the strength of the causal relationships between these events using human annotators.

Our resulting benchmark, **CRAB**, contains ∼2.7K high-quality event pairs, their causal score, and the respective documents in which the events appeared. All the events are grouped into 1.8K causal frames and 352 causal chains. We use this benchmark to assess the abilities of state-of-the-art (SoTA) models to understand and reason over the causal relationships of real-world events present in a set of contexts (*i.e.*, online news articles). Our analysis reveals that LLMs can capture explicit causal statements through pre-training, but they face difficulty applying causal reasoning to new scenarios, limiting their generalization and accuracy in offering predictions and explanations. We further stratify our results based on the structures of causal frames and chains, showing that they struggle with assessing the causality between events derived from complex causal structures compared to simple linear causal chains, especially when these events are extracted from different documents.

## Preliminaries on Causality

In this section, we define the main causality concepts that we use to create and analyze **CRAB**.

**Actual Causality**    Actual causality refers to the causal relationship between specific events and their causes in the real world (Halpern 2016) and seeks to understand the precise mechanisms by which one event leads to another, going beyond mere correlation. Research in causal inference has attempted to formalize actual causality using causal models that map how humans perceive and attribute cause and responsibility to events and their outcomes. However, human perception of causality usually depends on background context, implicit biases, epistemic state, and lack of information, making the task of actual causality attribution challenging to formalize (Matute et al. 2015; Henne et al. 2021). Additionally, in cases where the responsibility of an event can be attributed to more than one preceding event, observers tend to assign different attribution to the contributing causes (Wolff and Shepard 2013). Therefore, when events are described with natural language, the causal judgments are not binary but relative, enabling comparisons between causal events (Icard, Kominsky, and Knobe 2017).

**Causal Frames & Causal Chains**    Humans tend to attribute different degrees of causality between contributory events, relying primarily on domain and commonsense knowledge (Kıcıman et al. 2023). Causality research refers to this set of candidate events relevant to a particular outcome event as a Causal Frame (Halpern 2016). We construct **CRAB** to collect causal frame subgraphs, where each event is associated with its potential causes. Similarly, we explore the chain of events across time that leads to an outcome event E (Pearl 2009). We define the causal chain of an outcome event as the set of paths ending at E in

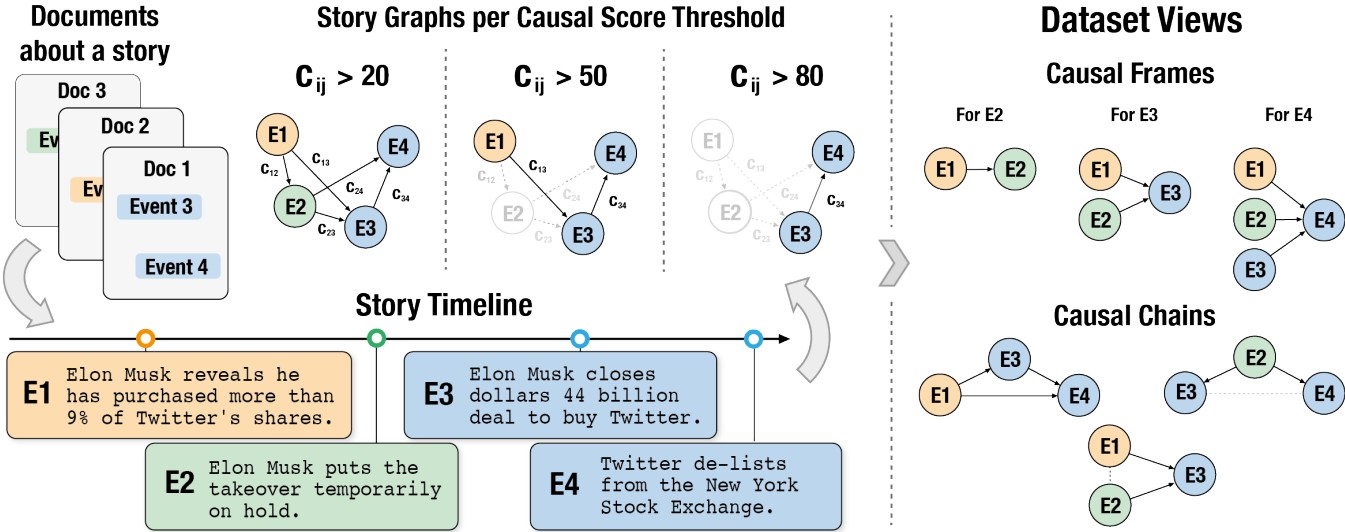

Figure 3: **CRAB** data pipeline overview: We collect documents covering newsworthy stories, create a timeline with the main events extracted from the documents for each story, and crowdsource human annotations on the causal judgments between the events *(score 0 to 100)*. Based on these scores, we generate a causal graph for each story that can be filtered on different causal score thresholds. **CRAB** can also be viewed from the perspective of causal frames and causal chains. Same-color events originate from the same document.

| Type of pairs | Pairwise Event Causality Score | | | |
|---|---|---|---|---|
| | Below 20 | 20-50 | 50-80 | Above 80 |
| **In-doc** | 3.9 | 25 | 26.6 | 44.4 |
| **Cross-doc** | 13.1 | 37.6 | 25.3 | 24 |
| **All pairs** | 11.9 | 35.9 | 25.5 | 26.7 |

Table 1: Percentage of pairs present in the **CRAB**, per causality score class.

the event's causal graph. **CRAB** leverages both concepts of causal chains and temporality, providing a testbed to assess the ability of language models to perform causal reasoning in different causal chain structures as depicted in Figure 2 (right).

## Dataset Construction

**Overview** We consider a set of documents covering a news story. Each document reports several events associated with that story. The time-ordered list of these extracted events defines a *timeline* associated with the story. From a timeline and its set of documents, we can build a *causal event graph*. Our goal is to identify the causal relations between the events in the graph using the documents in which these events are mentioned. The full data creation pipeline is described in Figure 3.

**Event Extraction** We select news articles from 20 stories about major events that happened around the world and we extract the main events mentioned in each document. In contrast to prior work (Shi and Lin 2019), we use a generative approach to extract the main events given a news piece.

Specifically, we prompt GPT-3 (text-davinci-003, Ouyang et al. 2022) to extract the main events from a given document (similar to Veyseh et al. 2021). Because generative methods come with the limitation of hallucinations and wrong outputs, we manually filter extracted events to keep only the valid generations. After filtering, our dataset contains 384 unique events. As causality is conditioned on temporality, we manually create a timeline of the extracted events for each of the 20 stories, by considering all documents associated with the story. While building these timelines, we disambiguate the events mentioned in different documents by merging differently phrased instances of the same event.

**Event Causality Linking** In the final stage of our pipeline, we collect causality judgments about all event pairs extracted from the documents related to a specific story (2730 pairs). Motivated by the way cognitive studies capture judgments about actual causality (Gerstenberg et al. 2021), we define the causality between real-world events not as a single binary score but as a continuous value from 0 to 100, enabling finer analysis and predictions of causal judgment. We qualify 44 Amazon Mechanical Turk workers, and for each pair of events, task 7 workers with providing a judgment for the causal link between the events (see Appendix for details regarding annotators' agreement scores).

## Dataset Analysis

**CRAB** consists of a set of 173 documents regarding 20 different stories discussing newsworthy real-world events. It contains 384 extracted unique event instances and 2730 event pairwise causality scores (see Tables 5 and 6 in Appendix for additional descriptive statistics). The experiments

| Tasks | Models | All pairs | In-doc | Cross-doc | Pre-Jan 2022 | Post-Jan 2022 |
|---|---|---|---|---|---|---|
| **Pairwise Causality Score** $C \xrightarrow{\text{H/M/L/N}} E$ | Flan-Alpaca | 21.6 | 14.9 | 22.4 | 22.0 | 21.3 |
| | GPT-3 | 25.8 | 24.4 | 25.4 | 26.6 | 25.2 |
| | GPT-4 | **54.7** | **59.0** | **53.7** | **56.4** | **53.5** |
| **Multi-class Pairwise Causality** $C \xrightarrow{\text{H/M/L/N}} E$ | Flan-Alpaca | 11.0 | 12.2 | 10.8 | 11.2 | 10.7 |
| | GPT-3 | 35.0 | 27.4 | 34.9 | 35.0 | 34.5 |
| | GPT-4 | **45.6** | **46.1** | **45.0** | **43.1** | **46.7** |
| **Binary Pairwise Causality** $C \xrightarrow{0/1} E$ | Flan-Alpaca | 60.1 | 73.8 | 56.7 | 62.1 | 58.7 |
| | GPT-3 | 57.2 | 67.0 | 55.0 | 56.9 | 57.5 |
| | GPT-4 | **73.9** | **80.0** | **72.6** | **76.5** | **72.0** |
| **Graded Causality** *(MCQ)* | Flan-Alpaca | 39.9 | 53.2 | 29.3 | 44.1 | 35.4 |
| | GPT-3 | **59.7** | **70.9** | **50.7** | **64.5** | **54.5** |
| | GPT-4 | 53.8 | 67.3 | 43.1 | 63.3 | 44.5 |

Table 2: Macro F1-scores on SoTA LLMs on all *Pairwise Causality Inference* tasks and the *Graded Causality Inference* MCQ task. For the MCQ task, we stratify the results for in-doc and cross-doc based on whether the effect & correct cause are extracted from the same document.

presented in the following section are based on these 4 classes reported in Table 1. We stratify the dataset and get the causal frame of each event and we categorize these sub-graphs based on the strength of causal scores between them (in-degree edges of the causal frame graph). Similarly, we extract causal chains from **CRAB** based on the three causal structures; *Mediation*, *Confounding*, and *Collision* (Pearl 2009), depicted in Figure 2 (right).

## Experimental Setup

To evaluate how language models reason about causality, we define different experimental frameworks covering various causality assessment scenarios, similar to Kıcıman et al. (2023). We investigate three tasks in ascending order of complexity in terms of causal structure: pairwise causality inference, graded causality inference, and causal chain inference. We use two decoder-only instruction-following API-based models, *GPT-3* (text-davinci-003, 175B size) and *GPT-4*, with the settings suggested by OpenAI (2023): a *temperature* of 0.3 and a *maximum length* of 256 tokens. We additionally test **CRAB** using *Flan-Alpaca-GPT4-XL* (Chia et al. 2023), an open-source 3B size encoder-decoder model fine-tuned on instruction-following datasets: FLAN (Long-pre et al. 2023) and GPT4-Alpaca.[1]

**Pairwise Causality Inference**  To evaluate Pairwise Causality Inference, we first prompt the model to generate a scalar causality score between two events given a context (the documents from which the events were extracted), mimicking the benchmark human annotations. We mapped the causality intervals to descriptions of different degrees of causality and augmented the prompt instructions with score ranges and their explanations. The 4 classes and definitions are as follows (i) *High causality*: a link between events that

---
[1]https://instruction-tuning-with-gpt-4.github.io/

are definitely causally related to each other (causal score above 80), (ii) *Medium causality*: a link between events that might be slightly causally related to each other (causal score between 50 and 80), (iii) *Low causality*: a link between events that have a little causal connection (causal score between 20 and 50), and (iv) *No causality*: independent events (causal score lower than 20). We compare it with the average annotators' score in **CRAB**. The full prompt can be found in Table 12 in Appendix. We then evaluate the model's answer by mapping the generated score to the four classes and computing the Macro-F1 score (**Pairwise Causality Score** in Table 2). We also experiment with binary and multi-class classification tasks (**Binary Pairwise Causality** and **Multi-class Pairwise Causality** in Table 2, respectively), prompting the model to output a causality class instead of a raw score. The prompts for these tasks can be found in Tables 9 and 10 in Appendix.

**Graded Causality Inference**  As previously described, one of the main concepts in actual causality is graded causation or responsibility (Halpern 2016), which is the relative degree to which an event causally contributes to an effect. Thus, we go beyond pairwise causality and prompt models to rank the events that contributed more to the effect. We create a Multiple Choice Question (MCQ) Answering task that asks the model to provide the most contributory to an effect cause among several events. We construct the dataset for the experiment by using the causal frames of each event and selecting 4 possible causes. We then ask the model to select, based on these 4 choices, the cause with the highest causality score (see the prompt in Table 11 in Appendix).

**Causal Chain Inference**  Pairwise causality provides a strong indication of the causal relationship between two events. However, these events are usually part of a larger chain of events with complex causal patterns. In this experiment, we consider not only the relations between the causes

| MODELS | All Frames | | SD | | D | | SC | | C | | M | | N | |
|---|---|---|---|---|---|---|---|---|---|---|---|---|---|---|
| | F1 | EM | F1 | EM | F1 | EM | F1 | EM | F1 | EM | F1 | EM | F1 | EM |
| Flan-Alpaca | 10.2 | 0.8 | 5.0 | 0.1 | 19.8 | 11.1 | 3.7 | 0.0 | 8.0 | 0.1 | 13.0 | 0.7 | 6.4 | 3.0 |
| GPT-3 | 28.8 | 3.3 | 29.1 | 4.1 | 30.5 | **16.6** | 31.9 | 3.3 | 31.0 | **4.1** | 29.0 | 0.0 | 19.3 | 9.1 |
| GPT-4 | **38.9** | **6.1** | **45.2** | **16.7** | **39.8** | 11.1 | **40.6** | **6.7** | **38.3** | 3.1 | **36.8** | **1.3** | **39.4** | **24.2** |

Table 3: Macro F1 and EM scores on SoTA LLMs for ***Graded Causality Inference*** stratified for different Causal Frame types. SD is for Strong Direct Causality; D is for Direct Causality; SC is for Strong Contributory Causality; C is for Contributory Causality; N is for No Causality; M is for Mixed Causality. Please refer to Figure 2 for detailed visualization of the different *Causal Frame* structures.

| MODELS | Mediation | | Confounding | | Collider | |
|---|---|---|---|---|---|---|
| | F1 | EM | F1 | EM | F1 | EM |
| Flan-Alpaca | **49.4** | **25.4** | 38.1 | 10.0 | **29.3** | 8.2 |
| GPT-3 | 40.2 | 10.6 | 38.1 | 9.7 | 28.0 | **9.7** |
| GPT-4 | 38.2 | 5.8 | **44.9** | **20.9** | 25.1 | 5.4 |

Table 4: Macro F1 and EM scores on SoTA LLMs for the ***Causal Chain Inference*** stratified in different Causal Chain structures.

and the effect in causal frames but also how the causes are related to one another. We stratify the results of the pairwise causality experiment based on the three causal chain structures we mentioned; *Mediation, Confounding,* and *Collider* (Figure 2 (right)). Similarly to the previous experiment, we extract causal chains that fit the three patterns and compute the F1-score and the Exact Match (EM) between ***CRAB***'s causal class annotations (4 classes) and the causal class predictions for each edge in the causal chains. We report each model's average scores in Table 4.

## Experimental Results

**Causal Discovery** Table 2 provides the pairwise causality inference scores for the three pairwise sub-tasks. All models perform poorly on ***CRAB***, with *GPT-4* showing a higher performance in most of the tasks compared to *GPT-3* and *Flan-Alpaca*. Models under-perform when assessing both Binary and Multi-class Pairwise Causality inference, especially when assessing medium and no causality. Further investigation of the results shows that in these cases of misclassification, the model tends to predict *high causality* instead of *medium causality*, and *medium causality* instead of *no causality*, demonstrating that models tend to hallucinate stronger causal relationships than humans perceive. Misclassified cases can be found in Figure 5.

**Assessing Responsibility** Going beyond pairwise causality and assessing whether LLMs can assign responsibility among potential causes of a specific event, we show that models fail to capture complex causal structures. Table 3 shows that models perform better when assessing the causality of graph structures that contain causal scores that are well-separated from each other. Additionally, from Table 4, we notice a common struggle among all models regarding the Collider case.

**Multi-document Causal Reasoning** In Table 2, we report results for causal score prediction for both event pairs that are found in the same document and event pairs found across documents. In all experimental settings, we see better performance for event pairs extracted from the same document (in-doc pairs). Based on these results, models tend to perform well in the causal discovery of in-doc event pairs since documents usually express causal relationships in an explicit way, likely because narrators seek to draw explicit causal links between events. Interestingly, in the MCQ setting, we find that *GPT-4* wrongly assigns the highest responsibility to events that belong in the same document 33% of the time, indicating that models themselves may be biased to prefer in-document causal links, even when humans identify a different causal link across multiple documents. This result suggests that models are able to capture causality when it is explicitly referred to in one context but struggle when it is implicitly inferred across documents.

**Memorization vs. Generalization** Due to the lack of transparency of closed-source GPT models, concerns arise regarding whether LLMs, pre-trained on extensive internet data, were subjected to the test set of benchmarks during their pre-training phase (Jacovi et al. 2023). We study how the performance of *GPT-3/4* varies when identifying causality for real-world events occurring pre-Jan 2022 and post-Jan 2022 (the official threshold date for their training data source). We observe a substantial drop in performance for graded causality and pairwise causality score, suggesting that the performance of models can be affected by knowledge of the events memorized during their pretraining stage.

## Related Work

There has been extensive research on introducing challenging causal benchmarks on commonsense causal reasoning (Mooij et al. 2016; Kalainathan and Goudet 2019; Bethard et al. 2008; Dalal, Buitelaar, and Arcan 2023), as well as assessing causal reasoning from the perspective of plausible alternatives and counterfactuals (Roemmele, Bejan, and Gordon 2011; Frohberg and Binder 2021; Srivastava et al. 2022; O'Neill, Quillien, and Henne 2022). Similar to our work, there have been attempts to create benchmarks that incorporate the cross-document setup (Welbl, Stenetorp, and Riedel 2018; Tu et al. 2019) and causal structures (Jin et al.

2023a), but not on real-world events. Additionally, existing studies investigate whether NLP models understand causality (Feder et al. 2022; Jin et al. 2023b; Zhang et al. 2023), providing methods to quantify the causal abilities of language models (Cao, Williamson, and Choi 2022; Yu, Li, and Wang 2019; Dalal, Buitelaar, and Arcan 2023). Another line of work studies the improvement of causal reasoning generation using instruction prompting (Kıcıman et al. 2023), Chain-of-Thought (CoT; Wei et al. 2022), and prompt augmentation (Schick et al. 2023). **CRAB** differs from prior work regarding the nature of its events, real-world events, and the type of causal reasoning that assesses LLMs, actual causality.

## Conclusions

This work introduces **CRAB**, a new causal reasoning benchmark that contains causality annotations for ∼2.7K pairs of real-world events. Using **CRAB**, we explore how LLMs assess the causal relationships between events when the causal signal comes from different contexts. Additionally, we assess LLMs performance in identifying and assessing complex causal structures. Our findings suggest that SoTA language models perform poorly in pairwise causal inference and responsibility assignment when events are spread across documents. Furthermore, this weak performance is amplified when LLMs must identify causal relationships in complex causal structures rather than simple linear chains.

## Acknowledgements

We thank Negar Foroutan, Reza Banaei, Deniz Bayazit, Beatriz Borges and Mete Ismayilzada for reading and providing comments on drafts of this paper. We also gratefully acknowledge the support of the Swiss National Science Foundation (No. 215390), Innosuisse (PFFS-21-29), the EPFL Science Seed Fund, the EPFL Center for Imaging, Sony Group Corporation, and the Allen Institute for AI.

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

# Appendix

## Dataset Construction Details

**News Article Selection** Based on a selection of 20 distinctive stories, we crawl the web and select the top 20 news articles per story. When extracting articles related to a story that happened many years before, we noticed that the retrieved articles also covered recent events that were loosely related to the story's main events. Therefore, we use a time window of 9 months when extracting the articles for each story to keep only articles that have been published around the time that the respective story happened.

| Causal frame | # frames | # pairs | In-doc | Cross-doc |
|:---:|:---:|:---:|:---:|:---:|
| **SD** | 24 | 85 | 25 | 60 |
| **D** | 18 | 70 | 22 | 48 |
| **SC** | 30 | 254 | 26 | 228 |
| **C** | 98 | 789 | 103 | 686 |
| **N** | 33 | 100 | 18 | 82 |
| **M** | 149 | 1386 | 156 | 1230 |

Table 5: Number of frames for different types of causal frames (see Figure 2) along with the number of event pairs additionally stratified for the in- and cross-doc cases. SD is for Strong Direct Causality; D is for Direct Causality; SC is for Strong Contributory Causality; C is for Contributory Causality; N is for No Causality; M is for Mixed Causality.

| Causal chain | # Chains |
|:---:|:---:|
| **Mediation** | 774 |
| **Confounding** | 924 |
| **Collider** | 115 |

Table 6: Number of chains for different types of causal chains (see Figure 2).

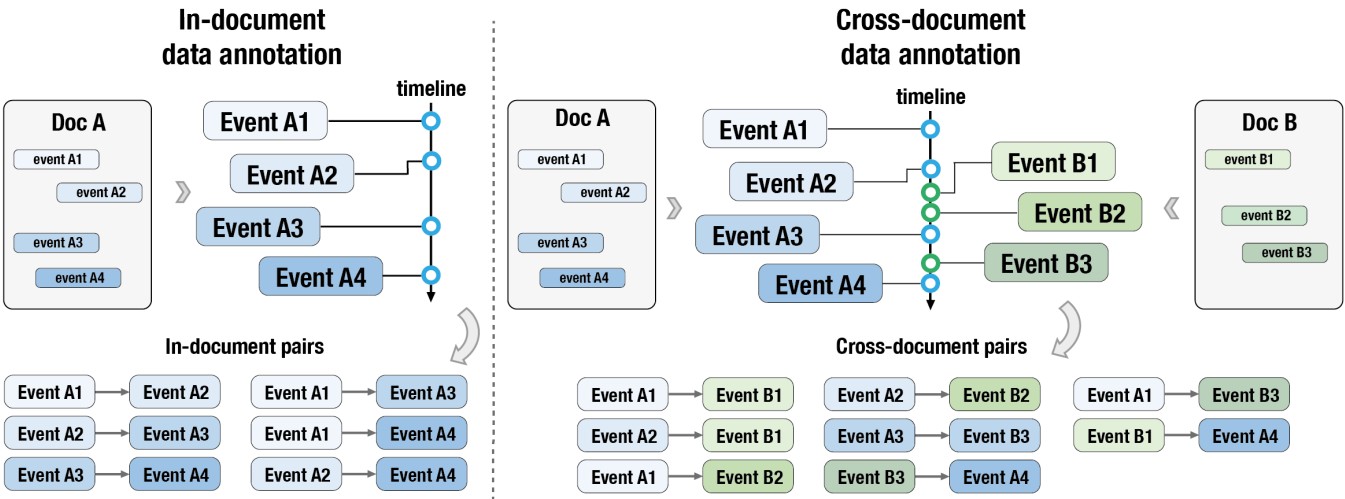

Figure 4: Dataset construction pipeline. Once events are extracted from the documents, we order them on time and formulate story timelines. Conditioned on the temporal ordering, we create all combinations of events in the document and pass the in-document pairs to annotators. We perform a similar process for the event pairs extracted from different documents (cross-doc), including an extra step of merging document timelines into one before taking the pair combinations.

**Event Extraction**   Using a generative approach for event extraction has two main benefits, confirmed through extensive experimental analysis. First, when prompted correctly, generative models successfully output structured information at the requested semantic abstraction, which leads to higher precision when extracting events. Second, the semantic granularity of the events we want to extract is between sentence and document level, meaning that we aim for the main events covered in the article and not syntactic events as existing works use (Ebner et al. 2020). Similarly to Smeros, Castillo, and Aberer (2019) and Romanou et al. (2020), we filter the top 20 news articles per story. We remove the ones with less than 100 words and those with paywalls or provide re-directions to the original referred news article. Finally, we clip the article to 250 words and round to the end of the sentence token. We end up using a total number of 384 news documents as the main test bed for event extraction.

**Crowdsourcing Causality Scores**   For the in-document annotations, we show the annotators the documents and ask them to assess 3 event pairs. For the cross-document ones, we give 2 documents at a time, along with 5 event pairs. Figure 4 depicts the pair creation process that served as input for Amazon Mechanical Turk experiments. To select native English speakers, we focus on the group of workers whose locations are in the USA.

We also ran a 2-phase qualification where we evaluated the quality of annotators on our task and selected the ones with a higher than 80% score on our qualification task. Finally, 44 out of 400 workers are selected as qualified. We pay each worker $0.80 for doing every 3 annotations for the in-document event pairs and $0.90 for doing 5 annotations for the cross-document pairs. Figures 6 and 7 depict the task instructions and annotation script used for crowdsourcing.

**Inter-rater agreement**   We have a total of 44 workers scoring the causality between 2730 pairs of events. All pairs are annotated by at least 7 annotators and, at most, 10 (around 21.3k annotations). We divide the causality score into 4 equal classes and compute Krippendorff's $\alpha$. We consider the ground truth as the majority vote. Table 8 shows Krippendorff's $\alpha$ for different groups of classes. As expected, the agreement to discriminate between the lowest causality class and the highest one is the highest, while it is harder for annotators to agree on discriminating between nearby classes. Krippendorff's $\alpha$ for all classes is 0.27, and 0.33 for the further classes. We note that the high number of annotators per sample increases the raw number of disagreements. Moreover, contrary to classical annotation situations where a small number of annotators label each sample, the Amazon Turk settings involve many different annotators participating in a task. Thus each sample is annotated by different workers, augmenting the variance and decreasing the agreement rate.

**Expert Annotation**   Given the low agreement, we select pairs where the average score falls on the boundary of the 4 classes and the variance between annotators is high to be validated by experts. This subset consists of 26.7% of the benchmark. This step is done by asking three expert annotators (NLP researchers who are familiar with the task of causal inference) further to annotate event pairs' causal scores and classes. Given the average causal score, the experts were asked to choose which of the neighboring classes was a better class for the event pair, updating the score accordingly. The inter-rater agreement, using Krippendorph's alpha, between experts is 0.70. These expert-validated causal scores and the remaining low-variance samples are used for **CRAB**.

| Causality Tasks | Model | Split by Date | Split by Story | Random split |
|---|---|---|---|---|
| **Pairwise Causality Score** | DeBERTa-large | 21.6 | 21.4 | 22.9 |
| | Llama2 7B | **24.3** | **26.3** | **32.8** |
| **Multi-class Pairwise Causality** | DeBERTa-large | **29.4** | **35.8** | **60.8** |
| | Llama2 7B | 23.2 | 23.1 | 32.7 |
| **Binary Pairwise Causality** | DeBERTa-large | **62.5** | **74.2** | **76.6** |
| | Llama2 7B | 51.1 | 51.9 | 58.5 |

Table 7: Macro F1-scores on test set for fine-tuned models on all Pairwise Causality Inference tasks. We stratify the results for Date, Story and Random splits. The best performance for each causality task is bolded.

| Causality classes | Size | $\alpha$ |
|---|---|---|
| $[1, 2]$ | 1310 | 0.04 |
| $[2, 3]$ | 1295 | 0.08 |
| $[3, 4]$ | 1429 | 0.12 |
| $[1, 4]$ | 1444 | 0.38 |
| $[1, 2, 3, 4]$ | 2730 | 0.28 |

Table 8: Krippendorff's $\alpha$ for different groups of classes.

would be interesting, and we hope that our paper provides a strong benchmark for pursuing this research direction.

## Experimental Results with Fine-Tuned LMs

We initially evaluated our proposed dataset on decoder-only models because decoder-only models (especially API-based ones such as GPT-3 / 3.5 / 4) have become important pillars of AI products, motivating researchers to benchmark their capabilities and identify their biases and limitations. However, it is important to additionally evaluate our causal benchmark on different architectures and inference techniques, providing additional insight into the difficulty of the task. On that note, we fine-tuned DeBERTa-v3-large (He, Gao, and Chen 2021) and Llama2-7B (Touvron et al. 2023) models.

We fine-tune both models on the 3 different pairwise causality tasks presented in our paper. For each task, we create 3 different train/test splits (75%/25% ratio) to study the generalization ability of the models after fine-tuning; Date: we select 5 out of the 20 most recent stories for the test set and the rest for the train, Story: we randomly select 5 stories for the test set and the rest for the train, and Random: we randomly split the event pairs dataset, regardless of the story or the date.

The results in Table 7 show that fine-tuned DeBERTa-large (encoder-only) models fail to perform well on CRAB, showing that our benchmark challenges the current state-of-the-art fine-tuned methods. Compared to decoder-only models in a few-shots setting, DeBERTa-large tends to underperform when splitting by story, except for the easier binary pairwise causality prediction task. Additionally, as expected, the experiments with the random data split have higher scores, which validate the information leakage of the context from the train to test set and verify that models rely on the context (articles) when assessing the causal relationship of the two events. A subsequent study on how fine-tuning improves pre-trained LLMs causal reasoning abilities

**PROMPT: Binary Pairwise Causality**

```
You are a helpful assistant for causal
relationship understanding.
Think about the cause-and-effect relationships
related to context.

Context:
<DOCUMENTS>

Event 1: <EVENT 1>
Event 2: <EVENT 2>
Did Event 1 cause Event 2 to happen?
Please answer in a single word: yes or no.
```

Table 9: Prompt for the ***Binary Pairwise Inference*** task.

**PROMPT: Graded Pairwise Causality - MCQ**

```
You are a helpful assistant for causal
relationship understanding.
Think about the cause-and-effect relationships
related to context.

Context:
<DOCUMENTS>

Event: <EFFECT>

What is the most likely cause of this event?
[A] <CAUSE 1>
[B] <CAUSE 2>
[C] <CAUSE 3>
[D] <CAUSE 4>

Let's work this out in a step-by-step way
to be sure that we have the right answer.
Then provide your final answer within the
tags, <Answer>A/B/C/D</Answer>.
```

Table 11: Prompt for the ***Graded Pairwise Inference*** task.

**PROMPT: Multi-class Pairwise Causality - 4 Classes**

```
You are a helpful assistant for causal
relationship understanding.
Think about the cause-and-effect relationships
related to context.

Context:
<DOCUMENTS>

Event 1: <EVENT 1>
Event 2: <EVENT 2>
How much did event 1 cause event 2 to happen?
[A] High causality: Event 1 is definitely
responsible for Event 2.
[B] Medium causality: Event 1 might have
been responsible for Event 2.
[C] Low causality: The context gives a
little indication that there is a connection
between the two events, but background info
might suggest a low causal connection.
[D] No causality: Events are somehow related
but definitely NOT causally related.

Let's work this out in a step-by-step way
to be sure that we have the right answer.
Then provide your final answer within the
tags, <Answer>A/B/C/D</Answer>.
```

Table 10: Prompt for the ***Multi-class Pairwise Inference*** task.

**PROMPT: Pairwise Causality Score**

```
You are a helpful assistant for causal
relationship understanding.
Think about the cause-and-effect relationships
related to context.

Context:
<DOCUMENTS>

Event 1: <EVENT 1>
Event 2: <EVENT 2>

What is the causality score between Event 1 and
Event 2 from 0 to 100?
Score above 80: Event 1 is definitely responsible
for Event 2.
Score between 50-80: Event 1 might have been
responsible for Event 2.
Score lower than 50 Events are somehow related
but definitely NOT causally related.

Let's work this out in a step-by-step way
to be sure that we have the right answer.
Then provide your final answer within the
tags, <Answer>score</Answer>.
```

Table 12: Prompt for the ***Pairwise Causality Score Inference*** task.

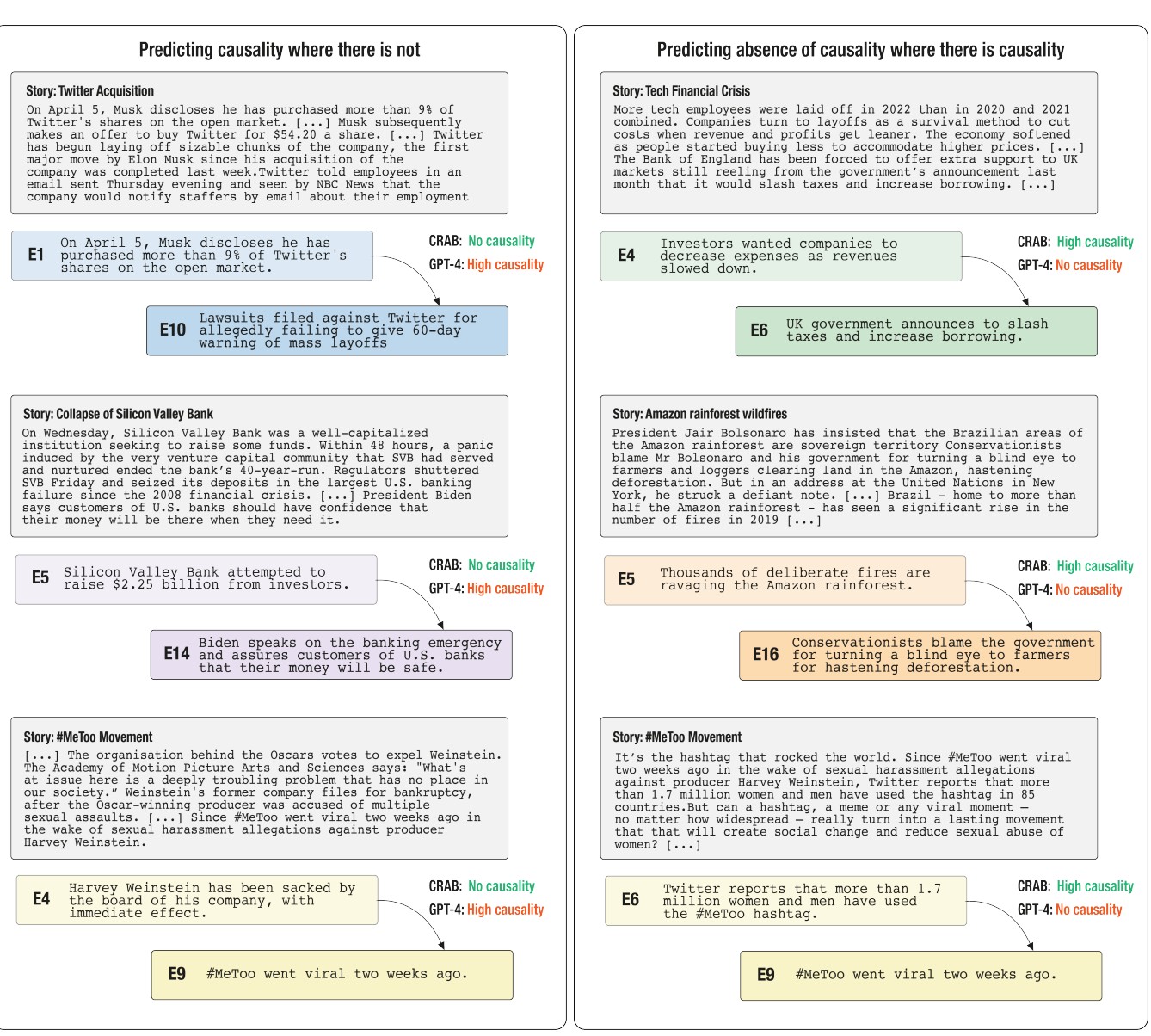

Figure 5: Failed predictions of GPT-4 regarding *CRAB* on high and no causal classes.

Thanks for participating in this HIT!

You will be given one segment of a news article story ( news piece ) that was published in the last ten years. You will be also given **3** pairs of events described in these news piece. The goal is to determine how much one event caused the other (**causal relationship**).

You will define the **causal relationship** (how much an event causes another one) between the events by choosing a **score** from **0** (No causal relationship) to **100** (strong causal relationship) **BASED ON THE NEWS PIECE**.

The score can be interpreted as:

- Score 81-100 : Event A is **definitely responsible** for Event B.
- Score 51-80 : Event A is **responsible** for Event B, **but not directly**: there are one or more intermediate events between Event A and B that could have led to Event B happening.
- Score 21-50 : The context doesn't give any clue about the connection between the two events; there might be a situation where there is a **small responsibility** of Event A to cause Event B.
- Score 0-20 : Events are somehow related but **definitely NOT causally related.**

The **EXAMPLE** below showcases different causality scores for various event pairs.

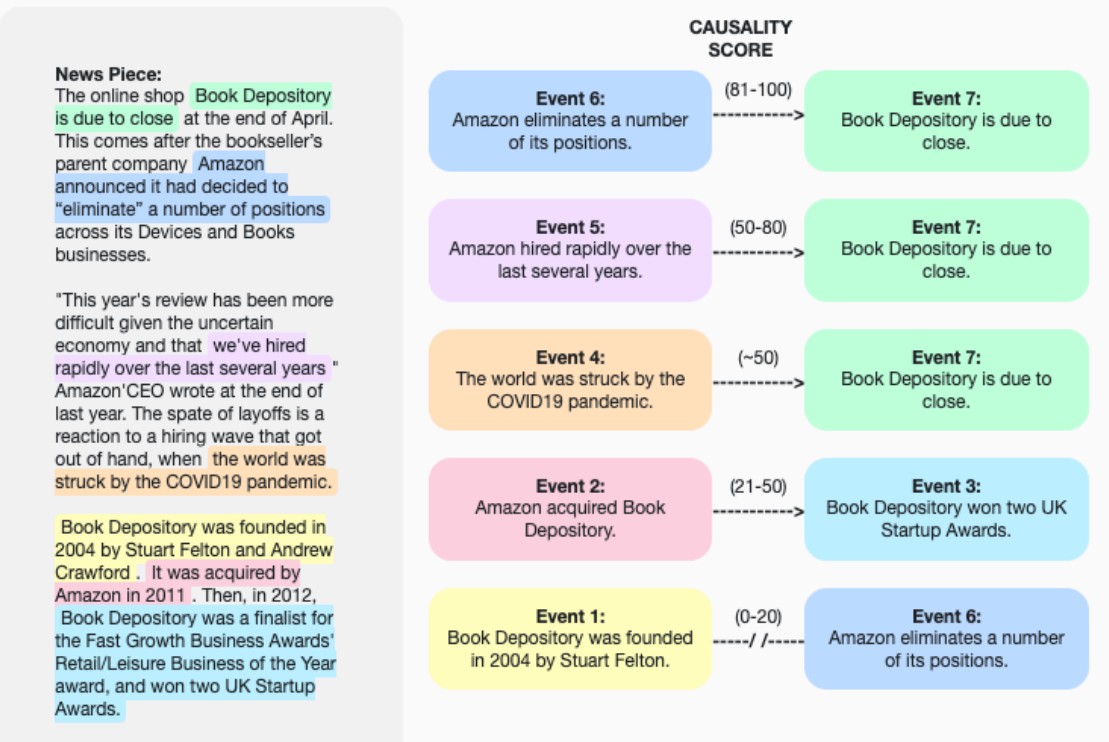

Along with the causality score, you will be asked **whether the second event would have happened if the first event didn't occur**. If your causal score in the previous question is relatively high, the second event probably couldn't have happened without the first one.

Figure 6: Amazon mTurk instructions script. Annotators need to perform the pairwise causality assessment task based on these instructions.

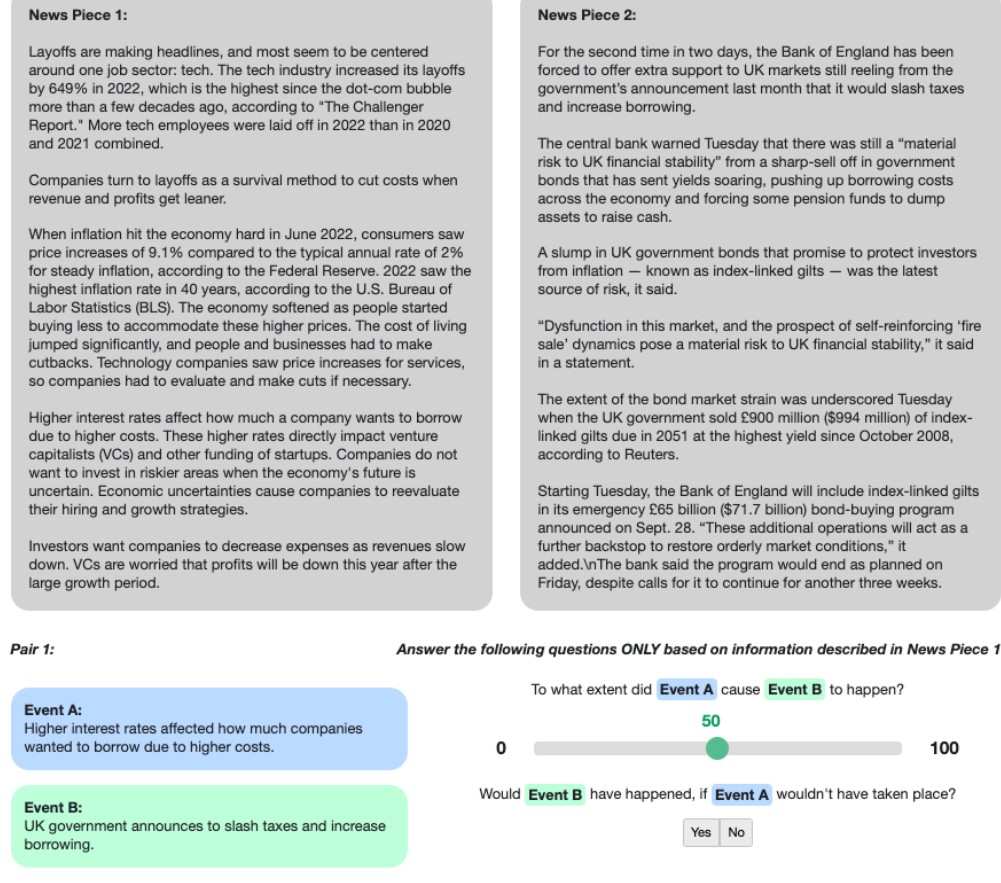

Figure 7: Amazon mTurk example script for annotating event pairs. Based on the instructions, annotators need to read the narratives and assess the causal relationship.