# OpenReview forum: "CRAB: Assessing the Strength of Causal Relationships Between Real-world Events"
_AAAI.org/2024/Workshop/LLM-CP — LLM-CP @ AAAI 2024 Oral_

### Official Review · Reviewer_5cQM · 2023-12-04
**Data set design and data acquisition seem to be sound. Minor flaws on annotations.**

**Rating:** 2
**Confidence:** 2

**Review:**

**Summary**

The authors present a novel dataset for benchmarking causal reasoning capabilities of language models. The benchmark is claimed to contain fine-grained contextual annotations of real world events of the last 10 years. LMs are used to extract relevant events from news texts, while annotation is performed by human annotators. The authors test multiple LMs on the newly acquired benchmark and claim poor performance on all tested systems.



**Novelty and Strengths**

The authors embed their work into the framework of actual causation. For evaluation, 6 possible types of frames for classifying the contribution of a particular event in relation to other events within a frame are considered. Judgment depends on the perceived relative strength of individual variables within a frame, allowing for a fine grained annotation of the causal relationships. To the best of my knowledge the authors are first to propose a natural language benchmark for probing the causal reasoning abilities of LMs with specific regard to actual causality on real world data.

1) Quality of the data set is ensured by manually checking, merging and sorting extracted events. While I'm no expert on the acquisition of annotation data via crowd-workers, I believe that the proposed methodology and obtained annotations of at least 7 workers are sufficient to guarantee the quality of annotations. The authors discuss agreement statistics of the obtained data in the appendix.
2) The models are evaluated extensively on four different types of tasks derived from annotation data. Each experiment seems to be well suited for measuring capabilities of LMs with regard to judging the strength of causal relations between events.



**Weaknesses**

1) The presented examples only consider positive contributing causal effects. The authors do not discuss inhibiting relations (compare e.g. to the discussion in "Embracing causality in default reasoning", Pearl 1988). The approach could be strengthened by including such considerations and discussing possible downsides or alternative types of causal frames under consideration.
2) The authors make use of 3 'expert annotators' to decide cases that lie near the border of two annotated classes. Since this seems to occur for 26% of the data and given the low number of experts, the paper could be strengthened by reanalyzing the expert annotated samples for possible biases.
3) In table table 12 the authors present a prompt for querying the LM. However, it is unclear how the score boundaries of the classes are chosen and how this choice would affect the predictions made by the LM. The paper could be improved by providing evaluations for other choices of boundaries (e.g. equally spaced class boundaries) and analyzing their effects on predictions.



**Minor**

* In their experiments the authors evaluate agreement of LMs with that of human judgements via annotated relations. While I agree that is it difficult to obtain objective annotations I would have expected a discussion on possible biases induced by human annotators; especially considering the fact that a larger part of the dataset was classified by only a small number of expert annotators.
* Word missing towards end of page two?: "[...] across time that [leads?] to an outcome even."

---

### Official Review · Reviewer_ZSum · 2023-12-06
**Benchmark for causal judgments about real-world events from narratives**

**Rating:** 2
**Confidence:** 3

**Review:**

The authors develop a causal reasoning benchmark that involves making causal judgments about real-world events described in narratives. The approach is interesting and I like the general idea of testing causal reasoning in more longitudinal, naturalistic contexts. However, I have a few concerns and questions about the methods:
- How was the set of causal frames in Figure 2 decided (I don’t believe they’re from Halpern 2016)? Are they exhaustive?
- How were the cutoffs for high vs. medium vs. low causality decided?
- The authors seem to advocate for aligning LLM causal judgments with those of humans, as they say that “SoTA language models perform poorly” because their F1 scores compared to annotators are low. But human causal judgments are susceptible to all kinds of biases, such as explaining away in collider structures. Do we also want LLMs to exhibit these biases? The authors should clarify their stance here and at least discuss some of the known biases in causal judgments from psychology.
- The task describes causal relationships in terms of both the extent to which A caused B, and how responsible A is for B. Causality and responsibility are related but different concepts, there’s quite a bit of work showing that the interpretation of responsibility varies a lot, especially when the events being asked about involve agents (e.g. see Vincent (2011). A Structured Taxonomy of Responsibility Concepts. for a review). This will lead to very different judgments of causality and responsibility. The experiment needs to be done more carefully here.
- The task also asks a counterfactual question — what was the purpose of this? Are these results reported anywhere?

---

### Official Review · Reviewer_zsov · 2023-12-06
**Well considered assessment method for causal relationships among real-world events**

**Rating:** 2
**Confidence:** 3

**Review:**

This paper presents a benchmark dataset and results for assessing the ability of LLMs to characterize a variety of token or actual causality relationships among real-world news events.

The assessment method and dataset are interesting and appropriate and well-focused for the LLM-CP workshop.

I appreciate the creation of the dataset, relying on recent news less likely to be in the LM training datasets; the consideration of causal connections across documents, and particularly the one-level-deeper analysis of how LLM's performance varies with linear and complex causal structures.

---

### Official Review · Reviewer_sbck · 2023-12-07

**Rating:** 3
**Confidence:** 3

**Review:**

This paper introduces CRAB, a benchmark designed to evaluate whether large language models' understand causal relationships. The authors measure the performance of several LLMs on CRAB and found that these systems achieve performance. This work show an important limitations of current LLMs, opening avenues for important future research.

---

### Meta-Review · Area_Chair_AyqD · 2023-12-14

**Recommendation:** 2
**Confidence:** 4

**Metareview:**

In this paper, the authors develop a new benchmark -- Causal Reasoning Assessment Benchmark -- for evaluating how well LLMs are aligned with the causal strength judgments of human annotators. They find that current LLMs are only poorly aligned with human judgments across a variety of tasks. This is interesting work and it received consistent ratings from the set of four reviewers. I recommend for it to be accepted at the workshop. One thing I wondered about was whether the online annotators were sufficiently incentivized to read the vignettes. As Figure 7 shows, these were fairly long, and maybe participants only looked tat the event pair without having read the full context? In future work, it would be nice to include a few examples for which there are normatively correct responses (based on the information provided in the scenario) to check that the annotators paid attention to the information. On a technical side, it would be interesting in future work to see to what extent LLMs can infer correct causal graphs from the stories, and whether prompting an LLM with a story + causal graph could lead to better aligned responses than provided either source of information by itself.

Some minor comments:

- In Figure 1, some of the links have arrows while others don't. It was a little counterintuitive that two independent events are in a box labeled "high causality".
- I didn't quite understand the visualizations of the causal frames in Figure 2. For example, why is the one in the middle on the bottom labeled "No causality" but then there is a link from C1 to E labeled "high".
- A collider is normally not classified asa a causal chain (so that's a little confusing in Figure 2).
- In Figure 7, the coloring is flipped between the items on the left and on the right (I hope that wasn't the case in the actual experiment as that would have been somewhat confusing).

---

### Decision · Program_Chairs · 2023-12-14

**Decision:**

Accept (Oral)

**Comment:**

Thank you for submitting your work to the LLM-CP workshop @ AAAI 2024. See you in Vancouver!